# Moisture Vapor Permeability and Thermal Wear Comfort of Ecofriendly Fiber-Embedded Woven Fabrics for High-Performance Clothing

**DOI:** 10.3390/ma14206205

**Published:** 2021-10-19

**Authors:** Hyun-Ah Kim

**Affiliations:** Korea Research Institute for Fashion Industry, 45-26, Palgong-ro, Dong-gu, Daegu 41028, Korea; ktufl@krifi.re.kr

**Keywords:** moisture vapor permeability, ecofriendly fiber, Tencel, bamboo, KES-F7, pore diameter

## Abstract

This study examined the moisture vapor permeability and thermal wear comfort of ecofriendly fiber-embedded woven fabrics in terms of the yarn structure and the constituent fiber characteristics according to two measuring methods. The moisture vapor permeability measured using the upright cup (CaCl_2_) method (JIS L 1099A-1) was primarily dependent on the hygroscopicity of the ecofriendly constituent fibers in the yarns and partly influenced by the pore size in the fabric because of the yarn structure. On the other hand, the moisture vapor resistance measured using the sweating guarded hot plate method (ISO 11092) was governed mainly by the fabric pore size and partly by the hygroscopicity of the constituent ecofriendly fibers. The difference between the two measuring methods was attributed to the different mechanisms in the measuring method. The thermal conductivity as a measure of the thermal wear comfort of the composite yarn fabrics was governed primarily by the pore size in the fabric and partly by the thermal characteristics of the constituent fibers in the yarns. Lastly, considering market applications, the Coolmax^®^/Tencel sheath/core fabric appears useful for winter warm feeling clothing because of its the good breathability with low thermal conductivity. The bamboo and Coolmax^®^/bamboo fabrics are suitable for summer clothing with a cool feel because of their high thermal conductivity with good breathability. Overall, ecofriendly fibers (bamboo and Tencel) are of practical use for marketing environmentallyfriendly high-performance clothing.

## 1. Introduction

The environmental impact of human beings has taken various forms, some familiar and others not generally recognized. The former includes energy consumption and pollution, together with global warming, melting icecaps, rising sea levels, and increasing frequency of adverse weather conditions [1] (p. 171). Albeit a minor contributor, the textile industry is exerting some impact, and the contribution of synthetic fibers such as PET and nylon must be taken into account. The term ecofriendly has been coined to define a process that is effective without harming the environment. Environmentally friendly fiber materials in textiles are divided into three areas: (organic) natural fibers, biodegradable synthetic fibers, and recycled fibers. An organic natural fiber implies organic cotton and bamboo, as well as Tencel as a regenerated cellulose fiber. Bamboo fibers made from bamboo pulp have a noncircular cross-section and impart good wear comfort with superior absorption and breathability while wearing clothing. It has 100% biodegradable characteristics (decaying after 3 or 4 months in the soil) and does not cause environmental pollution [2]. Tencel fibers developed by Lenzing AG in Austria originated from wood. The biobased Tencel (brand name of Lyocell) fibers are certified as compostable and biodegradable, and they can degrade after 3–4 months in the soil [2]. Biodegradable synthetic fibers include polylactic acid (PLA). PLA made from corn starch degrades after 3–5 years in the atmosphere and degrades after 2–4 months in landfill [2]. Recycled fibers are commercialized from recycled PET bottle and ECO CIRCLE^®^ developed by Teijin Fibers Limited in Japan. PET is manufactured from petrochemicals and will not decompose naturally. One manner in which this problem is tackled is through recycling [2]. The consumption of synthetic fibers such as PET, PP, and nylon is growing steadily every year with the appearance of highly functional new synthetic fibers. Two concerns in the textile industry are how the consumption of synthetic fibers can be reduced by substituting them with ecofriendly fibers with improved moisture transmission while wearing clothing, and how they can be achieved using various yarn manufacturing technology. 

Moisture transmission through textile materials is divided into two areas: moisture liquid transmission and moisture vapor transmission. Moisture liquid transmission involves a two-stage process: initially wetting and then wicking. In contrast, moisture vapor transmission is governed by the diffusion of moisture vapor through the inter-yarn and inter-fiber air spaces of the fabrics, called breathability. The moisture vapor transmission behavior of fabric materials plays a vital role in maintaining the clothing wear comfort. In particular, breathable fabrics that maintain high breathability with good perspiration absorption and fast-drying properties are needed in sportswear, work wear, and various types of protective clothing. In addition, high breathability in clothing allows the human body to provide cooling due to perspiration and evaporation. Moreover, minimizing sweat build-up in clothing is also important in cold environments. On the other hand, when examining the mechanism of moisture vapor transmission behavior from a human body wearing clothing, the first behavior is the diffusion of moisture vapor by sweating through the air spaces in the fabric. The drying of moisture and the moisture vapor absorbed by perspiration from the human body coincides, which is considered the second type of moisture vapor transmission behavior but is slightly complicated. 

Therefore, many studies [3,4,5,6,7] related to the moisture vapor permeability (MVP) of various fabrics have been carried out. They reported the breathable characteristics of fabrics according to the fiber materials and fabric structural parameters with various measuring methods of breathability. Rego et al. [3] examined the thermo-physiological wear comfort of cotton/polyester fabrics using a sweating guarded hot plate method. Gorjanc et al. [4] used a water cup measuring method to examine the effects of the fabric structural parameters on the thermal and moisture vapor resistance of cotton fabrics. Lee et al. [5] reported the effects of fiber materials and fabric structural parameters on the MVP using statistical modeling. Kim et al. [6] examined the relationship between the clothing performance of fabrics made from artificial and natural fibers and the dynamic moisture vapor transfer in a microclimate. Cubric et al. [7] explored the technical parameters affecting the moisture vapor resistance of knitted fabrics using a sweating guarded hot plate and a thermal manikin. Thus far, most studies have used traditional staple yarns and fabrics made from natural fibers and blended yarns using different methods to measure the MVP. 

On the other hand, some studies [8,9,10,11,12,13,14] on the moisture vapor transmission of water proof breathable fabrics have been performed under various conditions, such as steady-state, rainy, windy, and rainy and windy. In particular, Ruckman and coworkers [11,12,13,14] examined the condensation phenomena of waterproof breathable fabrics. Yoo and coworkers [15,16] analyzed the condensation of the inner surface of the fabrics at ambient temperatures below 0 °C in cold weather. The textile materials used in previous studies were divided into two areas: one on traditional fabrics using natural fibers, such as wool, cotton, and their blended fibers, and the other on waterproof breathable fabrics made from synthetic filaments, such as nylon and polyester. Moreover, the method for measuring breathability in each study was different, making it difficult to compare the breathability of waterproof breathable fabrics.

In particular, many studies [17,18,19,20,21,22,23,24,25,26,27,28,29] have examined breathability characteristics according to the measuring method. Lomax [17] reported a difference in the MVP between ISO 11092 and BS 7209 methods using coated nylon and PET fabrics. Salz [18] pointed out that the moisture vapor transmission rates are often difficult to compare because of various test methods. He developed a laboratory method for measuring the MVP using a heated cup method combined with an artificial rain condition. Yoo et al. [19] explored simulated heat and moisture transport characteristics in fabrics and garments determined using a vertical plated sweating skin model. Several studies [20,21,22,23,24,25,26,27,28,29] compared the performance of moisture vapor transmission of waterproof breathable fabrics using different measuring methods. Gibson et al. [20,21,22] examined the parameters affecting the steady-state heat and moisture vapor transmission measurements for clothing using a hydrophobic and hydrophilic membrane laminated with two- and three-layer fabrics. They used the most common techniques, such as a sweating guarded hot plate and the cup-type method to measure moisture vapor resistance and moisture vapor transmission rate.

Dolhan [23] reported a correlation between the upright cup and Canadian control dish methods when comparing the measuring apparatus for the moisture vapor resistance. Congalton [24] examined the heat and moisture transport of clothing ensembles and reported a strong correlation between the Hohenstein measuring method (ISO 11092) and the evaporative dish method (BS 7209), which was in contrast to Lomax [17], who reported an inverse correlation between the two measuring methods. Gretton et al. [25] reported a linear correlation between moisture vapor resistance measured using the Gore-modified desiccant method and the MVP index measured by BS 7209. McCullogh et al. [26] examined the correlation among five measuring methods of breathability using 26 waterproof breathable fabrics. They reported that the upright cup method showed the lowest water vapor transmission rate, followed by the dynamic cell method, inverted cup method, and desiccant inverted cup method. In addition, the correlation coefficient between the sweating guarded hot plate and desiccant inverted cup methods showed a high inverse correlation.

Huang [27] and Huang and coworkers [28,29] examined the factors affecting the moisture vapor resistance obtained using the ISO 11092 method and compared them with the existing water vapor transmittance method. Many studies carried out thus far have reported that the breathability of the fabrics differed according to the fiber materials, fabric structural factors, and surface modification method, such as coating and laminating, as well as the measuring method. Measuring the breathability of fabrics can be achieved using two methods. The first is to measure the water vapor transmittance rate (WVTR), which is a simple method used for quality control and marketing purposes. The second is to measure the moisture vapor resistance using the wet thermal resistance method, which is more precise (accurate) and used mainly for fabric development and research. On the other hand, many studies [30,31,32,33,34,35,36] focusing on improving wear comfort by sweating focused on the yarn and fabric manufacturing technologies by combining hydrophobic and hydrophilic yarns. Several wear comforts of woven fabrics made from various yarns were examined using a different yarn structure and various constituent fiber characteristics, such as Coolmax^®^, Tencel, Bamboo, and other ecofriendly fibers [37,38,39,40,41,42,43].

Despite these studies, there are few reports on the MVP (breathability) of the woven fabrics made from composite yarns, such as siro, siro-fil, and sheath/core yarns, using Coolmax^®^, Tencel, bamboo, PET, and polypropylene (PP) filaments. This study used bamboo and Tencel fibers to produce ecofriendly yarns with PET and PP filaments using siro and sheath/core spinning systems. The PET and PP filaments in the sheath/core or siro-fil yarn structures play a very important role in passing water and moisture vapor as drainage in the yarns and fabrics. Accordingly, PET and PP filaments were used to enhance the wear comfort characteristics with superior water absorption and moisture vapor permeability, even though they are not ecofriendly fibers.

There are no reports on the difference in breathability characteristics according to the yarn structure and the measuring method of breathability. Therefore, the main concern of this study is how the breathability of different composite yarn fabrics is influenced by the yarn structures and constituent fiber characteristics, and how it is associated with the thermal and absorption properties of the fabrics regarding the pore size of the fabrics according to the measuring method. Accordingly, in this study, two types of breathability, i.e., WVTR and moisture vapor resistance, by the wet thermal transmission of 15 fabric specimens made from different composite yarns were measured and compared in terms of the yarn structure and constituent fiber characteristics according to the two measuring methods. In addition, the moisture vapor transmission characteristics were compared with the thermal conductivity of the fabric specimens measured using the KES-F7 in terms of the pore size of the fabric and the thermal characteristics of constituent fibers.

## 2. Materials and Methods

### 2.1. Yarn Preparation

Coolmax^®^ (Dupont, Torrance, CA, USA) and Tencel (Gemeindeverwaltung, Lenzing, Austria) as ecofriendly fibers are used widely in the textile market. Various composite yarns have been made and commercialized using hi-multi PET and PP filaments in the functional and work wear market. In particular, a novel polypropylene (PP) filament with good absorption properties was applied to achieve clothing with thermo-physiological comfort [1,2,3]. On the other hand, few studies have investigated the moisture and heat transport of Tencel/Coolmax^®^-incorporated yarns and their fabrics according to the yarn structure with their measuring method. In this study, seven types of composite yarn were spun using ring, siro, and sheath/core spinning systems. Two existing hi-multi PET (75d/144f) and PP (100d/48f) filaments were used. Table 1 provides details of the yarn specimens. Sheath/core composite yarn specimens (no. (1), 147.5 dtex) were made by feeding PP DTY (draw-textured yarn) filament (30d/24f) between the double roving of Tencel. The siro-fil composite yarn (no. (2), 147.5 dtex) was prepared by replacing one of the siro components with a PET DTY filament (55d/216f) inserted at the back of the front rollers on the siro-spinning machine. The siro-spun yarn specimen (no. (3), 147.5 dtex) was made using Tencel roving on the siro-spinning frame. These were used as warp yarns. Sheath/core composite yarn specimen (no. (4), 196.7 dtex) was made by feeding Coolmax^®^ filament (50d/36f) between the double roving of Tencel. A ring-spun yarn specimen (no. (5), 196.7 dtex) was prepared using bamboo and Coolmax^®^ rovings on a ring spinning frame (Zinzer MAT 670, Krefeld, Germany).The siro-fil composite yarn specimen (no. (6), 196.7 dtex) was spun by replacing one of the siro components with a PET DTY filament (55d/216f) inserted atthe back of the front rollers on the siro-spinning frame. The bamboo spun yarn specimen (no. (7), 196.7 dtex) was made using bamboo roving. In addition, the hi-multi PET filament (no. (8), 83.3 dtex) and PP filament (no. (9), 111.1 dtex) were prepared as existing commercial yarns. These were used as weft yarns. Each warp yarn specimen was prepared as a 147.5 dtex and a 196.7 dtex for each weft yarn specimen by regulating the draft ratio on the ring (siro) and sheath/core spinning frames. Table 1 lists their twist multiplier (TM), spindle rpm, and blend ratio.

Table 2 presents a schematic diagram of the yarn specimens in this study, which was drawn with reference to the images of yarn cross-sections obtained by SEM and optical microscopy. As shown in Table 2, the yarn structure of the sheath/core composite yarns (no. (1) and (4)) was composed of filaments in the core and staple fibers in the sheath, respectively. The siro-fil yarns (no. (2) and (6)) were drawn as a side-by-side cross-section, which was assumed to be formed as two parts caused by the centrifugal force by the traveler rotation on the ring frame. The siro-spun yarn (no. (3)) was twisted using two Tencel rovings on the siro-spinning frame and had a compact yarn structure with uniformly distributed Tencel fibers in the yarn cross-section. The Coolmax^®^/bamboo spun yarn (no. (5)) was twisted by a traveler on the ring-spinning system and had a relatively bulky yarn structure with noncircular cross-sections of Coolmax^®^ and bamboo fibers. The bamboo spun yarn (no. (7)) had a slightly compact yarn structure with bamboo fibers distributed uniformly in the yarn cross-section. PET and PP filaments (no. (8) and (9)) were composed of parallel filament bundles and had yarn structures with many pores in the yarn cross-section.

### 2.2. Fabric Preparation

Fifteen types of fabric specimens were woven on a rapier loom (GTX 4-R, Picanol, Belgium), which was divided into three groups according to the warp beams. Three types of warp beams were prepared on a single warping machine (ROM2, Karl Mayer, Germany) using yarn specimens (1), (2), and (3) (Table 1), with which three groups of fabric specimens were woven using five types of weft yarn specimens ((4) to (8) in Table 1). Yarn specimen (9) was alternatively inserted as second weft yarn. Table 3 lists these 15 fabric specimens. Group A was composed of five different fabric specimens ((1) to (5)) using five weft yarn specimens ((4) to (8) in Table 1), with the first warp beam made from PP/Tencel core/sheath yarn (no. (1) in Table 1). Group B was composed of five types of fabric specimens ((6) to (10)) using the same five weft yarn specimens ((4) to (8) in Table 1) with the second warp beam made from PET/Tencel siro-fil yarn (no. (2) in Table 1). Group C was composed of five fabric specimens ((11) to (15)) using the same five weft yarn specimens ((4) to (8) in Table 1) with the third warp beam made from Tencel siro-spun yarn (no. (3) in Table 1). PP DTY (111.1 dtex/48f) was inserted alternatively for all fabric specimens as a second weft yarn for a plain weave pattern.

The warp density of all fabric specimens was 36.0 ends/cm and 24.6 picks/cm for the weft. The fabric weight was calculated using yarn linear density and fabric density of the fabric specimen. The fabric thickness was measured at a pressure of 2 gf/cm^2^ using a FAST-1 compression meter [42]. Fifteen types of gray fabric specimens, 20 m long each, were prepared and followed by dyeing and finishing processes. Gray fabric specimens were scoured on a CPB scouring machine (BPB, Kuester, Germany) and washed at a speed of 50 m/min on a continuous drying machine (Extra-CTA 2400, Benninger, Switzerland). A preset was done at 40 m/min at 150 °C. Dyeing was performed on a rapid dyeing machine (Cut-MF-1, Hisaka work Ltd., Osaka, Japan) at 120 °C for 60 min, followed by a drying treatment on a continuous dryer machine (Shrink dryer, Ilsung Ltd. Co., Seoul, Korea).The final setting was performed on a stenter machine (Sun-super, Ilsung Ltd. Co., Seoul, Korea) at a speed of 50 m/min at 130 °C.

### 2.3. Measurement of Pore Size of the Fabric Specimens

The moisture and heat transport of woven fabrics were strongly dependent on the constituent fiber characteristics, pore size, and fabric structural parameters [45,46,47,48,49]. Kim and Kim [50] reported that the thermal conductivity, drying rate, and air permeability of hollow filament-embedded woven fabrics were strongly dependent on the porosity and pore size of the fabrics. In this study, the primary concern of pore size measurements was how the pore size is influenced by the constituent yarn structure and how it affects the moisture vapor transport of fabrics according to the measuring method. The pore diameter (D, µm) was measured using a capillary flow porometer (CFP-1200 AE PMI Co., Ithaca, NY, USA) according to the ASTM measuring method. Figure 1 presents the porometer used in this study.

A fabric specimen 47 mm in diameter was placed in the specimen holder shown in Figure 1. The specimen holder was closed, and slight gas pressure was applied to eliminate the possible liquid backflow. The gas pressure was increased slowly. Finally, the lowest pressure at which a steady stream of bubbles rises from the central area of the liquid reservoir was recorded. The maximum pore diameter (D) was calculated using Equation (1) from the median value of the graph between the airflow and pressure.
D = C Ƴ/p,(1)
where D, Ƴ, and p are the maximum pore diameter (µm),the surface tension of the liquor (dynes/cm), and pressure (psi); C = 0.415 when p is in psi units. The yarn and fabric cross-sections were measured by field-emission scanning electron microscopy (FE-SEM, S-4100, Hitachi Co., Omori, Japan) and optical microscopy (i-Camscope-305A, Seoul, Korea).

### 2.4. Measurement of the WVTR of the Fabric Specimens

The resistance to moisture vapor diffusion (i.e., moisture vapor resistance) depends mainly on the air permeability of the fabric and indicates its ability to transfer perspiration vapor leaving of human skin. In this study, the main concerns of moisture vapor transmission measurements are how the moisture vapor resistance is associated with the absorption rate of the fabrics according to the constituent yarn structure (porosity) in the fabric, and how the thermal conductivity of the fabric is related to the moisture vapor resistance and is dependent on the yarn structure and thermal conductivity of the constituent fibers. The WVTR (g/m^2^·h) was measured using the JIS L 1099A-1, which was based on BS7209, similar to the upright cup method, as shown in Figure 2a. Five aluminum cups, 6 cm in diameter and 2.5 cm in height, were prepared, and the cup inside was heated to 40 °C by heating in an air-conditioned room (container), then filled with 33 g of CaCl_2_ as a desiccating agent. Five fabric specimens,7 cm in diameter, were prepared and conditioned at 20 ± 2 °C and an RH of 65 ± 2% for 24 h. The fabric specimen was laid 3 mm away from CaCl_2_ in an aluminum cup; its surface was faced toward the CaCl_2_ in the cup. Packing rubber with a circular covering was clamped and sealed over the fabric specimen on the mouth of the aluminum cup to prevent the leakage of moisture vapor between the fabric specimens and the aluminum cup. Five aluminum cup assemblies with fabric specimens were placed in the conditioning room at 40 ± 2 °C and 90 ± 5% RH for 1 h. The water vapor transmission rate was calculated using Equation (2).
WVTR (g/m^2^·h) = 10(W_2_ − W_1_)/(A × t),(2)
where, WVTR is water vapor transmission rate (g/m^2^∙h), W_2_ is the mass of the fabric specimen (mg) after the test, W_1_ is the mass of the fabric specimen (mg) before the test, A is the specimen area (cm^2^), and t is the testing time (h).

### 2.5. Measurement of Moisture Vapor Resistance of the Fabric Specimens

Moisture vapor resistance (R_ef_, m^2^Pa/W) of the fabric was measured using a sweating guarded hot plate (Therm DAC, London, UK) according to the ISO 11092 method. Figure 2b presents a schematic diagram of this apparatus. A fabric specimen, 50.8 cm × 50.8 cm in size, was prepared and conditioned in a standard atmosphere with an RH of 65% and a temperature of 20 °C. The specimen was placed over the PTFE membrane on perforated metal on a hot plate, which was used to prevent water on the perforated metal of the hot plate from wetting the fabric specimen. The temperatures of the guarded hot plate and air in the chamber were kept at 35 ± 0.5 °C (i.e., the temperature of human skin) with an RH of 40% and an air speed of 1 m/s. The moisture vapor resistance (R_ef_) of the fabric was determined by measuring the evaporative heat loss (H) under the steady-state condition, using Equations (3) and (4).
(3)$$R_{e,t}=\frac{\left({\;p}_{s}-p_{a}\;\right)A}{H},$$
where R_e,t_ is the total resistance to evaporative heat transfer provided by the fabric system and air layer (m^2^·Pa·W^−1^), A is the area of the plate test section (m^2^), p_s_ is the water vapor pressure at the plate surface (Pa), p_a_ is the moisture vapor pressure in the air (Pa), and H is the power input (heat loss)(W).
R_e,f_ = R_e,t_ − R_e,a_,(4)
where R_e,f_ is the resistance to evaporative heat transfer provided by the fabric (i.e., moisture vapor resistance of fabric), and R_e,a_ is the resistance measured for the air layer and liquid barrier. The arithmetic mean of five readings from each fabric specimen was calculated.

### 2.6. Measurement of the Thermal Conductivityof the Fabric Specimens

Thermal transmission through textile materials is divided into two methods: wet and dry heat transmission. Moisture vapor transmission by wet heat transport is governed by diffusion and convection, whereas dry heat transport occurs through conduction, convection, and radiation from the human body to the atmosphere. Moisture vapor resistance measurements using the ISO 11092 method were assessed using the principle of wet heat transmission, which means the movement of wet heat particles evaporated by perspiration sweated from human skin. In this study, one concern of the thermal transport measurement was how the wet heat transmission related to the moisture vapor resistance is associated with the dry heat transmission, and how they are influenced by the yarn structure and measuring method. Accordingly, the thermal conductivity of the fabric specimens as a measure of dry heat transport was measured to determine how it is influenced by the constituent yarn structure and then how it affects the moisture vapor resistance of the fabric specimens. The thermal conductivity of the fabric specimens was measured using the KES-F7 system (Kato Tech. Co., Ltd., Kyoto, Japan), of which a schematic diagram is shown in Figure 2c. First, the B.T. Box temperature was set to 30 °C, and water was circulated at a constant temperature of 20 °C in a water bath. A fabric specimen was placed on the water bath. Heat flowed from a high temperature (B.T. Box 30 °C) to a low temperature (water bath, 20 °C) in the apparatus through a plate and specimen. The B.T. Box (composed of an electrical system equipped with temperature sensors) then measured the heat loss emanating from the plate as watts (W) from the change in electrical voltage. The heat loss (W/10^−4^m^2^) with the fabric specimen placed on the water bath is H in Equation (5). The thermal conductivity (K) was calculated using Equation (5) as follows:(5)$$K=\frac{H}{t}\times \frac{D}{A\cdot T},$$
where, K, H, and D are the thermal conductivity (W/10^−2^ m·°C), dry heat loss (W/10^−4^ m^2^), and fabric thickness (10^−2^ m), respectively. A and t are the fabric area (10^−4^ m^2^) and time (h), respectively. ΔT is the temperature difference (°C).

### 2.7. Measurement of the Absorption Rateof the Fabric Specimens

The moisture vapor particles sweated from the human body move throughout the fabric and are partly adsorbed and wetted, after which drying will occur. The absorption rate of the fabric specimens was measured using a drying rate measuring apparatus (IT-ACD, INTEC Co. Ltd., Tokyo, Japan), as shown in Figure 3. A square fabric specimen (40 cm × 40 cm) was conditioned at 20 °C and 65% RH in the conditioning room (JIS L 1096, 2010), and the initial mass (m_1_) was then measured. A square fabric specimen was submerged in distilled water for 20 min at 27 ± 2 °C in a water bath. The specimen in the distilled water bath was passed through a mangle at 25 cm/s, and the mass (m_2_) was weighed. The absorption rate (%) of each fabric specimen was calculated using Equation (6).
(6)$$R\;\left(\%\right)=\left(m_{2}-m_{1}\right)/m_{1},$$
where R is the absorption rate of fabric (%), m_1_ is the initial mass (g) of the fabric specimens, and m_2_ is fabric mass (g) after passing through the mangle.

## 3. Results and Discussion

### 3.1. Pore Size of the Fabric Specimens with SEM Images of the Cross-Sections of Yarns

The fabric porosity affecting clothing wear comfort is divided into two types: micro and macro porosity [51,52]. In particular, the moisture vapor and heat permeabilities are incorporated with both micro and macro porosities. In this study, the calculated porosity does not apply because the fabric specimens prepared in this study were made from the same yarn count and fabric sett, which means that the calculated porosity may not be available to examine the difference in the MVP among the fabric specimens prepared using the same yarn count and fabric sett. Therefore, in this study, the pore diameter was considered a measure to examine the WVP according to the yarn structure and measuring method. Table 3 lists the measured physical properties of the fabric specimens with the measured pore diameters. ANOVA (F-test) was carried out to verify the statistical significance of the experimental data shown in Table 4. ANOVA was performed between the mean value of the physical properties of each specimen (five specimens) in each group with the 95% confidence limit (5% significance level). Table 5 lists the ANOVA analysis of the physical properties of 15 fabric specimens. As shown in Table 5, the significance test between each mean pore diameter among the five specimens in each group A, B, and C was statistically significant, as F_0_ (V/Ve) > F (4, 20, 0.95) and *p* < 0.05. Similarly, WVTR, R_ef_, and thermal conductivity were statistically significant, as F_0_ (V/Ve) > F (4, 20, 0.95) and *p* < 0.05, as shown in Table 5. Figure 4 presents a diagram of the pore diameters (mean) with the deviation of the fabric specimens listed in Table 4. The deviation in Table 4 denotes the difference between maximum and minimum values of five experimental data of each specimen.

As shown in Figure 4, of the three fabric specimen groups (A, B, and C), the pore diameters of the fabric specimens in group A ((1) to (5)) showed higher values than those in groups B ((6) to (10)) and C ((11) to (15)). This suggests that the pore size of the PP/Tencel sheath/core yarns used as a warp yarn of group A was larger than that of the PET/Tencel siro-fil and Tencel siro-spun yarns used as the warp yarns of groups B and C. Hence, the sheath/core yarn has larger pores and voids. By contrast, the siro-fil and siro-spun yarns have compact yarn structures, resulting in relatively small pore diameters in the yarns, even though the fabric specimens were made from the same yarn count and fabric density. These were verified by SEM and optical microscopy of the constituent warp yarns used in the fabric specimens. Table 6 presents SEM and optical microscopy images of the yarn specimens shown in Table 1. As shown in yarn specimen (1), which was used as a warp yarn of fabric specimens (1) to (5) (group A), many air voids were observed in the sheath and core. Round-shaped capillary channels at the border between the Tencel fibers and the filaments in the core of the PP/Tencel sheath/core yarn were found, resulting in a higher pore diameter of fabric group A ((1) to (5)) than fabric groups B ((6) to (10)) and C ((11) to (15)). On the other hand, as shown in yarn specimen (2) in Table 6, a compact yarn cross-section was observed in the PET/Tencel siro-fil yarn, resulting in a smaller pore diameter of the fabric specimen group B ((6) to (10)). In yarn specimen (3) in Table 6, small air voids were observed in the Tencel siro-spun yarn, resulting in relatively small pore diameters of the fabric specimen group C ((11) to (15)). Regarding the pore size of the fabric specimens according to the yarn structure in the weft direction, of the five types of fabric specimens ((1) to (5), (6) to (10), and (11) to (15)), as shown in Figure 4, the Coolmax^®^/Tencel sheath/core fabric (specimen (1)) and Coolmax^®^/bamboo spun fabric (specimen (2)) exhibited larger pore diameters than the PET/Tencel siro-fil fabric (specimen (3)). By contrast, the pore diameter of the siro-fil fabric (specimen (3)) was smaller than that of the bamboo spun fabric (specimen (4)) and hi-multi PET filament fabric (specimen (5)). These results suggest that sheath/core and spun yarn fabrics have larger pores in the fabrics, whereas siro-fil has a compact yarn structure, which results in small pores in the fabric. Relatively large air voids in the Coolmax^®^/bamboo spun yarns observed in yarn specimen ((5)) in Table 6 were noted in the weft direction in fabric specimen (2), resulting in relatively large pore diameters for fabric specimens (2), (7), and (12), as shown in Table 4 and Figure 4. On the other hand, a compact yarn cross-section was observed in yarn specimen (6) in Table 6, resulting in small pore diameters for fabric specimens (3), (8), and (13), as shown in Table 4 and Figure 4. Similar to yarn specimen (5), yarn specimen (7) showed a relatively compact yarn cross-section, as shown in Table 6. The pore sizes in yarn specimen (7) were smaller than those of yarn specimen (5), resulting in smaller pore diameters of fabric specimens (4), (9), and (14) than those of fabric specimens (2), (7), and (12), as shown in Table 4 and Figure 4. Regarding the hi-multi PET filament of the yarn specimen (8), many air voids were observed in the SEM and optical microscopy images of the yarn cross-section. In addition, non-twisted parallel filament bundles were observed in the SEM image of the yarn surface shown in Table 6, which produced fine capillary channels along the filament bundles with many pores in the yarn cross-section, as shown in the SEM and optical microscopy images of yarn specimen (8) in Table 6, resulting in a high pore diameter of fabric specimens (5), (10), and (15) compared to siro-fil and spun yarn fabrics made from yarn specimens (5), (6), and (7) in Table 6. According to previous studies [45,46], the MVP is incorporated with micro and macro porosities, and fine voids cause microporosity among the fibers in the yarns. By contrast, macro porosity is produced from the void spaces among the threads in the fabric. The pore diameters measured in this study were assessed as a measure of the fabric porosity considering both micro voids among the fibers in the yarns and macro voids among the threads in the fabrics. Therefore, the MVP was examined in relation to the pore diameter measured from SEM images of the air void and capillary channels formed according to the different yarn structures. In addition, the MVP of the fabrics made from the different yarn structures was compared and discussed with its two methods (WVTR and R_ef_) for measuring the breathability, as shown in the next section.

### 3.2. WVTR of the Fabric Specimens Using Upright Cup Method

Figure 5 presents the WVTR of the 15 fabric specimens. The mean value of the five specimens for the WVTR in groups A, B, and C was statistically significant, as shown in Table 5. The mean values of the 15 specimens were plotted with the maximum and minimum values of the five experimental data of each specimen, respectively. A comparison of the WVTR according to the warp yarn structure (i.e., groups A, B, and C) revealed the WVTR of group A to be higher than that of groups B and C because of the larger pore diameter by the warp yarn (PP/Tencel sheath/core) of group A than groups B (PET/Tencel siro-fil) and C (Tencel siro-spun), as shown in Figure 4. This was verified by SEM images (Table 6), i.e., larger pores with a capillary channel between PP filament in core and Tencel fibers in the sheath were observed in the yarn specimen (1) (a warp yarn of group A fabric specimens), which resulted in a higher WVTR of group A fabrics. The WVTR of group C was slightly higher than group B because of the larger pore diameter by the warp yarn (Tencel siro-spun) of group C than group B (PET/Tencel siro-fil), as shown in Figure 4. These results were consistent with Fohr et al. [54], who reported that the WVTR was strongly dependent on the porosity and diffusion characteristics of the moisture vapor particles, which is in agreement with the current findings.

The WVTR of the fabric specimens was examined according to the yarn structure in the weft direction. As shown in Figure 5, of five fabric specimens ((1) to (5)) in group A, the WVTR of fabric specimen (1) was higher than that of specimens (2) and (3). That of fabric specimen (2) was slightly higher than that of specimen (3), because of the larger pore diameter by the weft yarn (Coolmax/Tencel sheath/core yarn) of fabric specimen (1) than fabric specimen (2) (Coolmax/bamboo spun yarn) and 3 (PET/Tencel siro-fil yarn), and of fabric specimen (2) than (3). Similar results were shown in groups B ((6) to (10)) and C ((11) to (15)). On the other hand, the WVTR of fabric specimen (4) (as well as (9) and (14)) inserted with bamboo spun yarn was the highest compared to the other fabric specimens, which was attributed to the high absorption rate and the diffusivity of bamboo spun yarn fabric with the appropriate pores in the yarns. As shown in Table 3, of the 15 fabric specimens, specimens (4), (9), and (14) composed of bamboo spun yarns in the weft direction exhibited the highest absorption rate, which was partly responsible for the highest WVTR of the fabrics. These results are in accordance with previous studies [55,56] reporting that the water vapor transmission of hygroscopic fibrous materials was higher than that of the materials that do not absorb moisture. This reduces the moisture built up in the microclimate, enhancing moisture vapor transmission from human skin to the environment. According to Li et al. [57], an increase in the WVTR by the absorption of moisture vapor is mainly because the heat of sorption increases the temperature of the fibrous assemblies, which in turn affects the moisture vapor transmission rate. The effect of water vapor absorption on the WVTR can be explained in fabric specimens (5), (10), and (15). As shown in Figure 5, the WVTR of fabric specimens (5), (10), and (15) showed the lowest value compared to the other specimens. This is because the PET 75d/144f filaments inserted in fabric specimens (5), (10), and (15) are hydrophobic and do not absorb moisture, resulting in a low WVTR. Summarizing the WVTR measured by the upright cup method according to the yarn structures in the warp and weft directions, the WVTR of the fabric specimens divided into groups A, B, and C was dependent on the pore diameter of the fabric, i.e., the WVTR of group A fabrics (specimens (1) to (5)) was higher than that of group B (specimens (6) to (10)) and C (specimens (11) to (15)) fabrics, which was attributed to the larger pore diameter of the group A fabrics. On the other hand, a study of the WVTR of fabric specimens according to yarn structure in the weft direction revealed the WVTR to be primarily dependent on the absorption rate of the constituent fibers in the fabric and partly on the pore size of the fabric. The fabric specimens composed of bamboo fibers with hygroscopic characteristics in the weft direction (i.e., high absorption rate) exhibited the highest WVTR. In contrast, the fabric specimens composed of hydrophobic PET filaments showed the lowest WVTR. Hence, in this study, the WVTR of the fabric measured using the upright cup method according to the weft yarn structure and fiber characteristics was strongly dependent on the hygroscopicity of the constituent fibers. On the other hand, the WVTR of the fabric according to the warp yarn structure was governed by the pore diameter of the fabric, i.e., dependent on the warp yarn structure.

### 3.3. Moisture Vapor Resistance of the Fabric Specimens by ISO 11092 Method

The ISO 11092 method uses a sweating guarded hot plate apparatus to simulate moisture transport through the textile when worn next to the human skin. This model measures the moisture vapor resistance of the fabric by measuring the evaporative heat loss in the steady state. Its measuring mechanism is different from the upright cup method. Figure 6 shows the moisture vapor resistance (R_ef_) of the fabric specimens. The mean value of the moisture vapor resistance of the 15 fabric specimens was statistically significant, as shown in Table 5.

The moisture vapor resistance of the 15 fabric specimens according to the weft yarn structure showed a distinctive result, and a similar trend among fabric groups A, B, and C was observed, i.e., proportional to the pore diameters. As shown in Figure 6, fabric specimens (1) and (5) in group A, (6) and (10) in group B, and (11) and (15) in group C exhibited low moisture vapor resistance, i.e., superior breathability to other fabric specimens. These results were attributed to the larger pore sizes (specimens (1), (5), (6), (10), (11), and (15) in Figure 4) in the fabrics depending on the weft yarn structure. Hence, the superior moisture vapor transmittance of fabric specimens (1), (6), and (11) was due to the following: high pore diameters with the capillary channels (yarn specimen (4) in Table 6) between the Coolmax^®^ noncircular filaments in the core and Tencel fibers in the sheath of the yarns, and the fine capillary channels (yarn specimen (8) in Table 6) between the hi-multi PET filaments of fabric specimens (5), (10), and (15), which enable more moisture vapor to be transmitted from the fabric toward the outside. On the other hand, fabric specimens (3), (8), and (13) showed the highest moisture vapor resistance, i.e., inferior breathability to that of the other fabric specimens. This was attributed to the low pore diameters (specimens (3), (8), and (13) in Figure 4) in the fabrics because of the compact yarn structure of the PET/Tencel siro-fil yarns (yarn specimen (6) in Table 6), which prevents moisture vapor from being pushed by the moisture vapor pressure. In addition, the moisture vapor resistance of the fabric specimens composed of spun yarns in the weft direction ((2) and (4) in group A, (7) and (9) in group B, and (12) and (14) in group C) was higher, i.e., showed inferior breathability to that of the fabric specimens with the sheath/core and hi-multi PET filament. This was attributed to the smaller pore diameters (Figure 4) of the spun yarn fabrics than the sheath/core and hi-multi PET fabrics and partly to the higher hygroscopicity of the bamboo fibers. According to previous studies [1,58], the correlation between the diffusion process and moisture vapor transmission can be explained by the swelling of the fibers due to the affinity of the hydrophilic fiber molecules. Hence, as moisture vapor diffuses through the fibers in the fabric, it is absorbed by the fibers, causing fiber swelling and reducing the size of the air (void) spaces between the fibers. This delays the diffusion process, which reduces the rate of moisture vapor particle movements [58]. This explains why hygroscopic spun yarn fabrics (specimens (2) and (4) in group A, (7) and (9) in group B, and (12) and (14) in group C) exhibited higher moisture vapor resistance than the filament fabrics (specimens (5), (10), and (15)). This explains why the R_ef_ measured by the sweating guarded hot plate method differs from the WVTR measured by the upright cup method, which is due to the difference of mechanism between the two measuring methods, i.e., the transmission of moisture vapor by forced convection due to CaCl_2_ in the upright cup and the diffusion process by the free convection of wet heat particles in a sweating guarded hot plate apparatus. Furthermore, these results indicate that the sweating guarded hot plate method is appropriate to measure the breathability of coated (or laminated) nylon (or PET) fabrics, whereas the upright cup method is suitable for non coated ordinary fabrics. 

### 3.4. Thermal Conductivity of the Fabric Specimens

The sweating guarded hot plate (ISO 11092 method) applies the transmission (diffusion, or convection) of wet thermal particles to measure the moisture vapor resistance, which is similar to the transmission by the conduction of dry thermal particles. Understanding how the moisture vapor resistance of the fabrics measured using the diffusion of wet thermal particles (sweating) is associated with the thermal resistance (conductivity) measured by the conduction of dry thermal particles is very important for examining dry and wet thermal wear comforts and their correlations according to the yarn structure of the fabric and the thermal characteristics of constituent fibers. Figure 7 presents the thermal conductivity of 15 fabric specimens. The mean value of the thermal conductivity of the 15 fabric specimens was statistically significant, as shown in Table 5.

As shown in Figure 7, the thermal conductivity of fabric specimens ((3), (8), and (13)) with PET/Tencel siro-fil in the weft direction was the highest, followed by the fabric specimens ((4), (9), and (14)) composed of bamboo spun yarns. These fabrics have smaller pore diameters (Figure 4) than the other fabrics. Small air voids in the low porosity fabric due to the compact yarn structure of the PET/Tencel siro-fil and bamboo spun yarns cannot entrap the neighboring air in the compact yarns and their fabrics. This enables easy heat conduction from the inner layer to the outer one of the fabrics, resulting in higher thermal conductivity than the other fabrics. On the other hand, fabric specimens (5), (10), and (15) in groups A, B, and C composed of the hi-multi PET in the weft direction showed the lowest thermal conductivity because of the larger pore diameter in the filament bundles (Figure 4), which entraps the neighboring air and prevents heat flow from the inner layer to the outer layer of the fabrics, resulting in lower thermal conductivity than the other fabrics. The lower thermal conductivity of these fabrics was attributed partly to the lower thermal conductivity of the PET filament than that of the Tencel and bamboo fibers (p. 150) [59,60]. In addition, the thermal conductivity of the Coolmax^®^/Tencel sheath/core fabrics (specimens (1), (6), and (11)) was lower than that of the Coolmax^®^/bamboo spun, bamboo spun, and PET/Tencel siro-fil fabrics (specimens (2) to (4), (7) to (9), and (12) to (14)) in groups A, B, and C, which was also attributed to the larger pore diameters of the sheath/core fabrics than the other fabrics. In particular, fabric specimens (1), (6), and (11) with Coolmax^®^/Tencel sheath/core yarns in the weft direction exhibited higher thermal conductivity than the hi-multi PET fabric specimens (5), (10), and (15), despite the sheath/core fabrics having much larger pore diameters than the hi-multi PET fabrics. The higher thermal conductivity of the Tencel fibers than the PET filament was considered the cause [59,60], because it may compensate for the thermal conductivity of the two fabrics, resulting in higher thermal conductivity of the Coolmax^®^/Tencel sheath/core fabrics. Overall, the thermal conductivity of the composite yarn fabrics is influenced mainly by the pore size according to the yarn structure and partly by the thermal characteristics of the constituent fibers. These results are in agreement with previous findings [61,62,63]. Das et al. [61,62] reported that cotton/acrylic blend fabric exhibited low thermal conductivity because of its high porosity. Das and Istiaque [63] published a similar result for the hollow yarn fabric. The thermal properties of the composite yarn fabrics were influenced mainly by the fabric porosity; the sheath/core yarn fabric with a noncircular cross-sectional filament exhibited low thermal conductivity. Furthermore, the bulky yarn fabric with noncircular cross-sectional fibers showed low thermal conductivity because of the high porosity and hairy and crimpy constituent fibers. Thus, the thermal conductivity of the composite yarn fabrics was governed partly by the thermal characteristics of the constituent fibers. Finally, considering the market application of ecofriendly fiber-embedded fabrics for high-performance clothing with good wear comfort, the Coolmax^®^/Tencel sheath/core fabrics are useful for winter clothing with a warm feel due to good breathability with low thermal conductivity. Bamboo and Coolmax^®^/bamboo spun yarn fabrics are suitable for summer clothing with a cool feeling because of their high thermal conductivity and good breathability.

### 3.5. Correlation Analysis between Wear Comfort Characteristics of Fabrics

Heat and moisture vapor transmission is essential for characterizing the thermo-physiological behavior of fabrics to assess their wear comfort for the design of high-performance clothing. They are characterized by two parameters: resistance to dry heat (inverse of thermal conductivity) and resistance to moisture vapor. Therefore, correlation analysis was carried out to determine the interrelationships between the thermal conductivity and moisture vapor resistance according to the yarn structure and constituent fiber characteristics with the measuring method, as well as the fabric absorption rate. Table 7 lists the correlation coefficient between each parameter. The correlation between the moisture vapor resistance (R_ef_) and pore diameter was significant at the 99% confidence level. In addition, the correlation between R_ef_ and thermal conductivity (K) was significant at the 99% confidence level. Therefore, the results obtained with significant correlation at the 99% confidence level were graphed. A trend line was added to the graph shown in Figure 8.

Figure 8 presents a diagram showing the interrelationships among the moisture vapor resistance, thermal conductivity, and pore diameter of the fabric specimens.

As shown in Table 7 and Figure 8a correlation coefficient between the moisture vapor resistance (R_ef_) and pore diameter was observed as an inverse correlation (−0.73). Hence, the moisture vapor resistance is influenced by the pore diameters of the fabrics and partly by the hygroscopicity of the constituent fibers, which is consistent with previous results [54,55,56,57,58]. Fohr et al. [54] and Lomax [64] reported that moisture vapor through a textile medium diffuses in two ways: through the air space (pore) and along the fiber itself. In addition, the diffusion rate along the textile material depends on the porosity, and the moisture vapor diffusivity of the fiber is affected by the absorption and its hygroscopicity. Barnes et al. [55] and Hong et al. [56] reported that the moisture vapor transmission of the hygroscopic fibrous materials was higher than that of non-absorbing materials. On the other hand, the correlation between R_ef_ and WVTR was very low (0.26) because the measuring mechanisms of the two methods were different. The upright cup method by CaCl_2_ measures the moisture vapor transport by Fick’s law, whereas R_ef_ measures the wet thermal transport by the difference in vapor pressure. This is in accordance with McCullogh et al. [26], who reported that the correlations between R_ef_ and the upright and inverted cup methods of WWB (waterproof, windproof, and breathable) shell fabrics made of nylon and PET were very low (−0.59 and 0.1, respectively). The results are in contrast to those by Huang and Qian [29] and Gorjanc et al. [4]. Huang and Qian [29] reported that the correlation coefficients between R_ef_ and the upright and inverted cup methods using ordinary and breathable fabrics were −0.87 and −0.66, respectively. Gorjanc et al. [4] reported that the correlation coefficient between the water cup method and moisture vapor resistance by the Permetest method using cotton and cotton-stretch fabrics was >0.9. Previous findings [4,26,29] suggested that the breathability characteristics according to the constituent yarn structure were dependent on the measuring method. In addition, the correlation coefficient between R_ef_ and thermal conductivity (K) was 0.87, meaning that the K values measured by dry heat transmission and R_ef_ measured by wet heat movement are governed by a similar mechanism: the movement of heat and moisture vapor particles, as shown in Figure 8b. Moreover, they are dependent on the pore size of the fabrics and the hygroscopicity and thermal characteristics of the constituent fibers.

## 4. Conclusions

The MVP and thermal wear comfort of the ecofriendly fiber-embedded woven fabrics were examined according to the measuring method in relation to their absorption and thermal properties with the constituent yarn structure and fiber characteristics. Fifteen fabric specimens composed of sheath/core, siro-fil, siro-spun, and ring-spun yarns were prepared using bamboo and Tencel as ecofriendly fibers, as well as PP, PET, and Coolmax^®^ as core filaments. The WVTR and moisture vapor resistance (R_ef_) were measured using the upright cup method by CaCl_2_ and a sweating guarded hot plate method, respectively. The WVTR measured by the upright cup method was dependent primarily on the hygroscopicity of the eco-friendly constituent fibers (bamboo and Tencel) in the yarns and partly influenced by the pore size in the fabric depending on the yarn structure. Of the 15 fabric specimens, the bamboo spun yarn fabrics exhibited superior WVTR, followed by the Coolmax^®^/Tencel sheath/core fabrics.

On the other hand, the hi-multi PET fabrics showed inferior WVTR, followed by the PET/Tencel siro-fil fabrics. The moisture vapor resistance (R_ef_) by the sweating guarded hot plate method was governed mainly by the pore size in the fabric and partly by the hygroscopicity of the constituent ecofriendly fibers. These results suggest that ecofriendly fibers, bamboo, and Tencel can contribute to environmental improvement and wear comfort related to water and moisture vapor transmission. The moisture vapor resistance of hi-multi PET filament fabrics was the lowest, i.e., best, followed by the Coolmax^®^/Tencel sheath/core fabrics. In contrast, the PET/Tencel siro-fil fabric was the highest, i.e., showed inferior breathability. R_ef_ measured by the sweating guarded hot plate method differed from the WVTR measured by the upright cup method due to the difference in measuring mechanism between the two methods.

The thermal conductivity of the composite yarn fabrics was influenced by pore size in the fabric and the thermal characteristics of the constituent ecofriendly fibers in the yarns. The hi-multi PET filament fabrics exhibited the lowest thermal conductivity, followed by the Coolmax^®^/Tencel sheath/core fabrics, whereas the PET/Tencel siro-fil fabrics showed the highest thermal conductivity, followed by the bamboo spun yarn fabrics. These results were verified by correlation analysis. The correlation coefficient between the moisture vapor resistance and pore diameter was −0.73. The correlation coefficients between the moisture vapor resistance (R_ef_) and thermal conductivity (K) and between K and pore diameter were 0.87 and −0.55, respectively. On the other hand, the correlation coefficient between the WVTR and moisture vapor resistance was very low (0.27), which was attributed to the different mechanisms between the two measuring methods, i.e., transmission of moisture vapor by forced convection in an upright cup and the diffusion process viathe free convection of wet heat particles in a sweating guarded hot plate apparatus.

Lastly, considering the market application for high-performance ecofriendly clothing with good wear comfort, the Coolmax^®^/Tencel sheath/core yarn fabrics are useful for winter clothing with a warm feeling due to the good breathability with low thermal conductivity. Bamboo and Coolmax^®^/bamboo spun yarn fabrics are suitable for summer clothing with a cool feeling because of their high thermal conductivity and good breathability. Although based on the MVP and thermal wear comfort obtained in this study, the market application of Tencel fibers for winter outdoor clothing and bamboo fibers for summer outdoor clothing is of practical use for engineering high-performance fabrics. These results suggest that an increase in the consumption of ecofriendly fibers with a decrease in synthetic fibers can reduce environmental pollution in the textile industry. 

## Figures and Tables

**Figure 1 materials-14-06205-f001:**
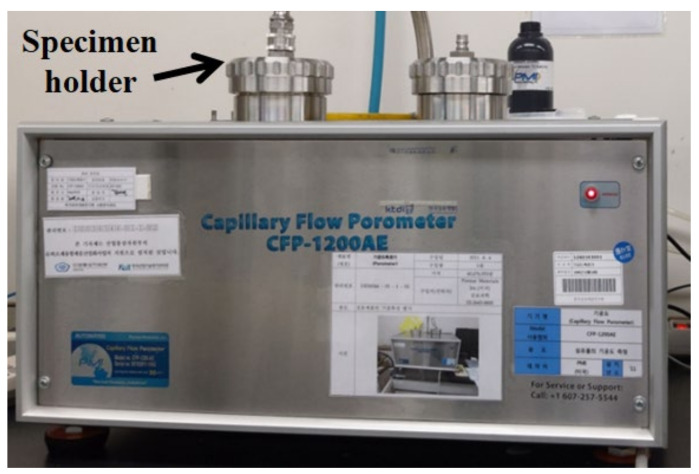
Capillary flow porometer machine.

**Figure 2 materials-14-06205-f002:**
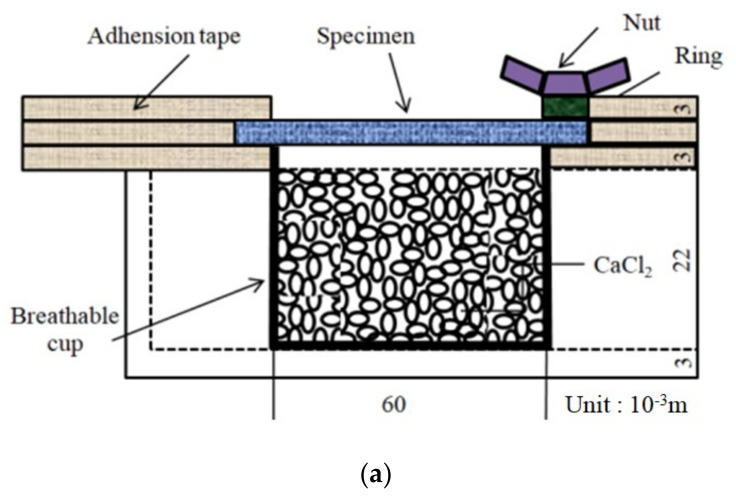
Schematic diagram of the measuring equipment of (**a**) upright cup method (JIS L 1099A-1), (**b**) sweating guarded hot plate method (ISO 11092), and (**c**) KES-F7 system [44].

**Figure 3 materials-14-06205-f003:**
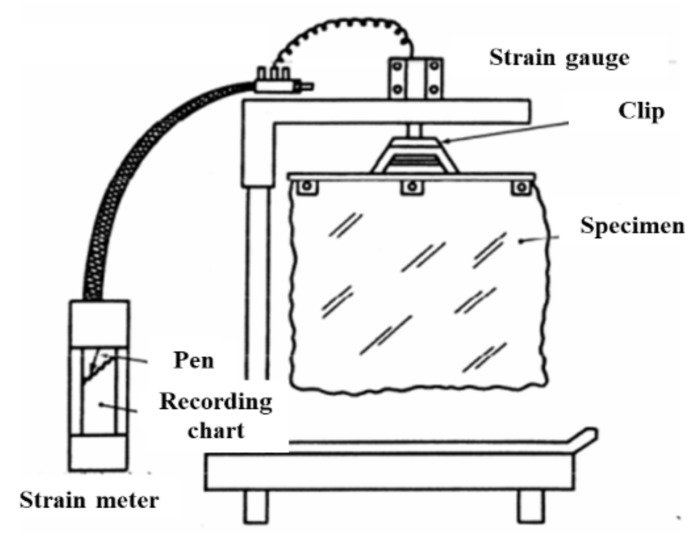
Schematic diagram of measuring equipment of the absorption rate.

**Figure 4 materials-14-06205-f004:**
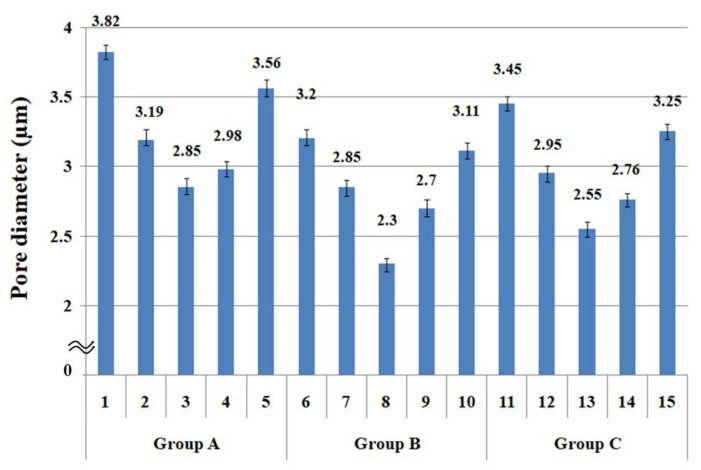
Diagram of pore diameters of the fabric specimens.

**Figure 5 materials-14-06205-f005:**
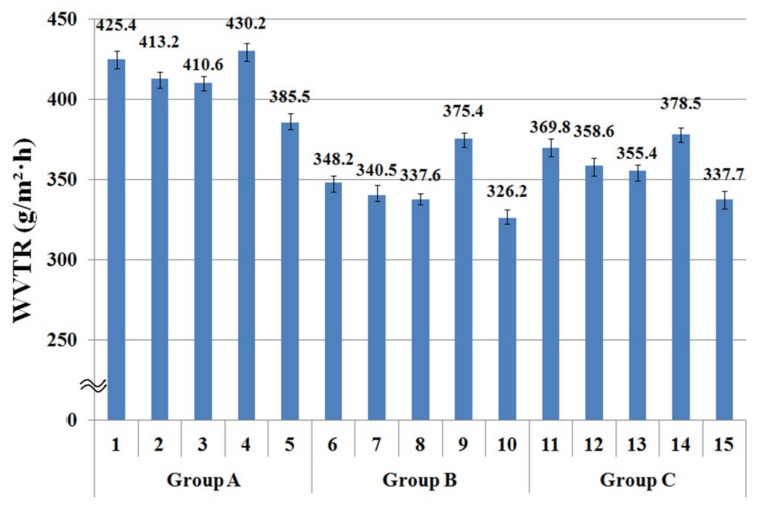
WVTR of the specimens using the upright cup method.

**Figure 6 materials-14-06205-f006:**
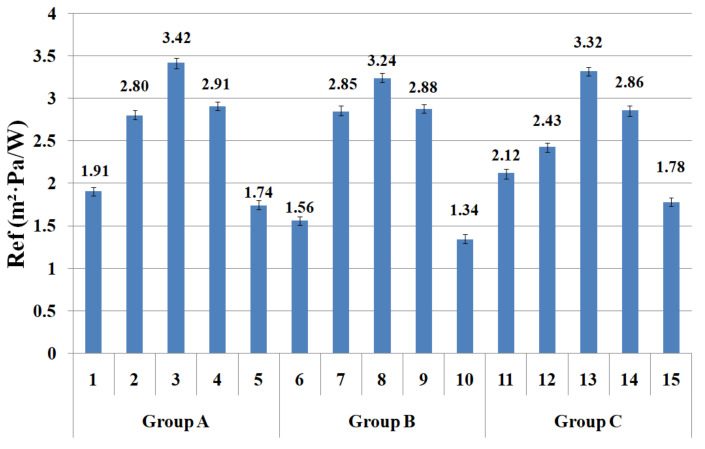
R_ef_ of the specimens using the ISO 11092 method.

**Figure 7 materials-14-06205-f007:**
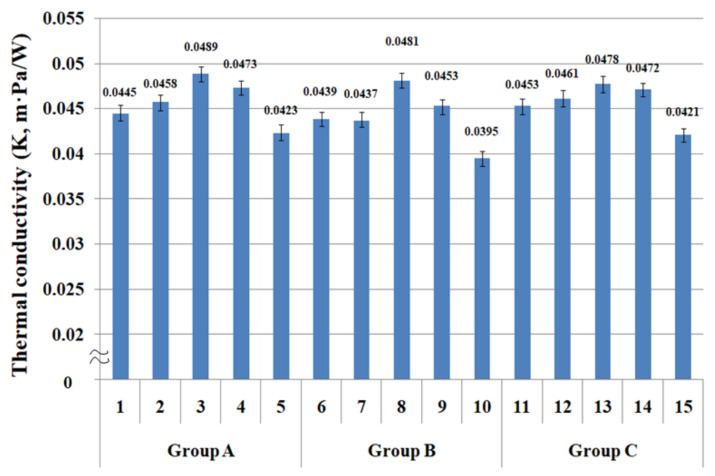
Thermal conductivity of the fabric specimens.

**Figure 8 materials-14-06205-f008:**
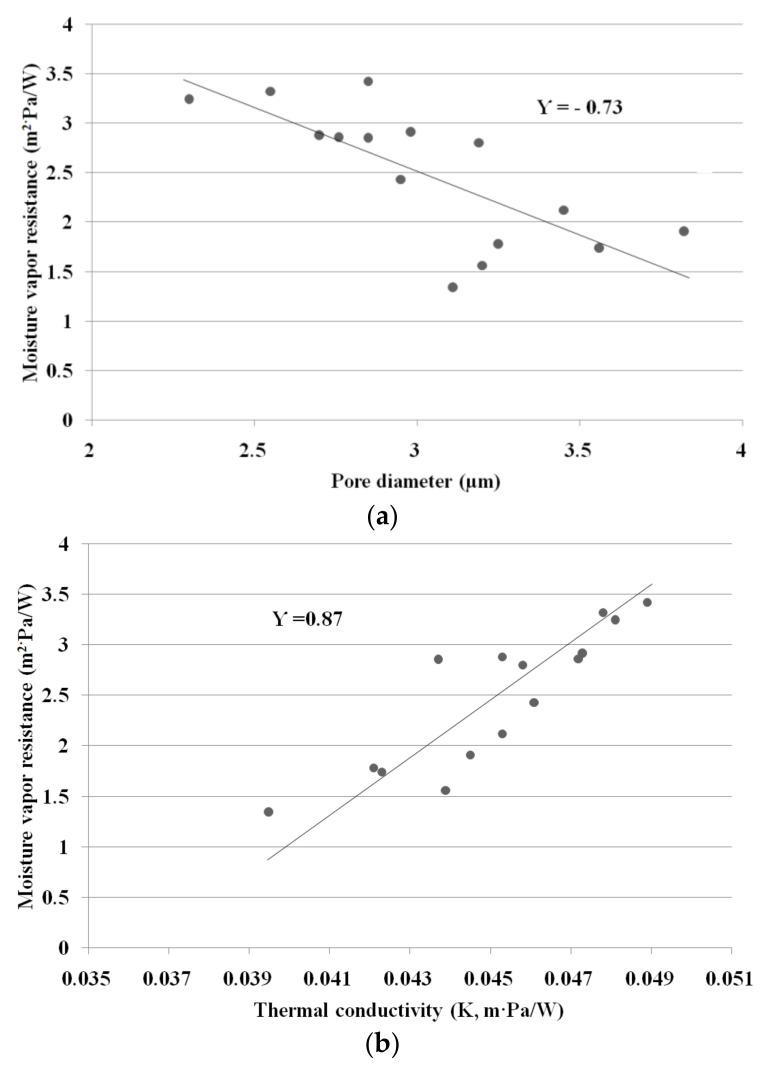
Correlation diagram between each wear comfort characteristic of the fabrics: (**a**) moisture vapor resistance and pore diameter; (**b**) moisture vapor resistance and thermal conductivity.

**Table 1 materials-14-06205-t001:** Details of the yarn specimens used in this study.

Yarn No.	Yarn Type	BlendRatio (%)	Spinning Method	Yarn No. ( dtex)	Twist		Fiber (Filament) Used	Period of Biological Decay in Soil (Month)
					TM tpi/$\sqrt{Ne}$)	Spindle (rpm)		
(1)	PP/Tencel S/C	PP: 39.3/ T: 60.7	Sheath/core	147.5	4.53	7000	PP DTY 30d/24f and Tencel S/F	PP: no decay T: 3-4
(2)	PET/Tencel Siro-fil	P: 44.4/ T: 55.6	Siro-fil	147.5	4.12	9000	PET DTY 55d/216f and Tencel S/F	P: no decay T: 3-4
(3)	Tencel Siro-spun	T: 100	Siro-spun	147.5	4.42	11,000	Tencel S/F	T: 3-4
(4)	Coolmax/TencelS/C	C: 39.3 T: 60.7	Sheath/core	196.7	4.34	9000	Coolmax 50d/36f and Tencel S/F	C: no decay T: 3-4
(5)	Coolmax/ BambooSpun	C: 48.6/ B: 51.4	Ring-spun	196.7	3.82	12,000	Coolmax/bamboo S/F	C: no decay B: 3-4
(6)	PET/Tencel Siro-fil	P: 44.4/ T:55.6	Siro-fil	196.7	4.12	9000	PET DTY 55d-216f and Tencel S/F	P: no decay T: 3-4
(7)	Bamboo spun	B: 100	Ring-spun	196.7	3.82	12,000	Bamboo S/F	B: 3-4
(8)	* Hi-multi PET	P: 100	-	83.3	-	-	PET DTY 75d/144f	no decay
(9)	* PP filament	PP: 100	-	111.1	-	-	PP DTY 100d/48f	no decay

Note: S/F: staple fiber, T: Tencel, *: existing filament. PP: polypropylene, C: Coolmax. P: PET, B: bamboo, DTY: draw textured yarn.

**Table 2 materials-14-06205-t002:** Composite warp and weft yarn models [44].

Model	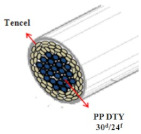	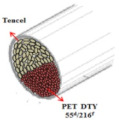	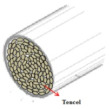
Spec.	PP DTY 30d/24f + Tencel sheath/core yarn	PET DTY 55d/216f + Tencel siro-fil for yarn	Tencel + Tencel staple fibres siro-spun yarns
Yarn specimens	PP/Tencel S/C 147.5 dtex No (1)	PET/Tencel siro-fil 147.5 dtex No (2)	Tencel siro-spun 147.5 dtex No (3)
Model	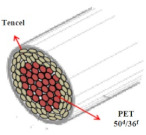	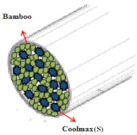	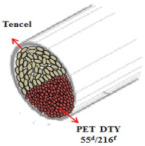
Spec.	Coolmax 50d/36f+ Tencel sheath/core yarn	Coolmax/bamboo staple fibers ring-spun yarns	PET DTY 55d/216f + Tencel siro-fil yarn
Yarn specimens	Coolmax/Tencel S/C 196.7 dtex No (4)	Coolmax/bamboo spun yarn 196.7 dtex No (5)	PET/Tencel siro-fil 196.7 dtex No (6)
Model	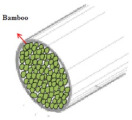	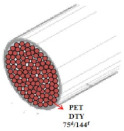	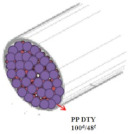
Spec.	bamboo staple fibees ring-spun yarns	PET DTY 75d/144f filament	PP DTY 100d/48f
Yarn specimens	bamboo spun yarn 196.7 dtex No (7)	Hi-multi PET 83.3 dtex No (8)	PP 111.1 dtex No (9)

**Table 3 materials-14-06205-t003:** Specification of the fabric specimens.

Group	Fabric Specimen No.	Warp Yarn	Weft Yarn		Fabric Density (Ends, Picks/cm)		Weight (g/y)	Thickness (10^−3^ m)
			Yarn 1	Yarn 2	Wp	Wf		
A	1	PP/Tencel Sheath/core (147.5 dtex)	Coolmax/Tencel S/C (196.7 dtex)	PP (111.1 dtex)	36.0	24.6	162	0.368
	2		Coolmax/bamboo spun (196.7 dtex)				162	0.345
	3		PET/Tencel Siro-fil (196.7 dtex)				162	0.341
	4		Bamboo spun (196.7 dtex)				162	0.364
	5		Hi-multi PET (83.3 dtex)				137	0.352
B	6	PET/Tencel Siro-fil (147.5 dtex)	Coolmax/Tencel S/C (196.7 dtex)	PP (111.1 dtex)	36.0	24.6	160	0.396
	7		Coolmax/bamboo spun (196.7 dtex)				161	0.337
	8		PET Tencel Siro-fil (196.7 dtex)				161	0.345
	9		Bamboo spun (196.7 dtex)				161	0.345
	10		Hi-multi PET (83.3 dtex)				137	0.294
C	11	Tencel Siro-spun (147.5 dtex)	Coolmax/Tencel S/C (196.7 dtex)	PP (111.1 dtex)	36.0	24.6	158	0.380
	12		Coolmax/bamboo spun (196.7 dtex)				160	0.356
	13		PET Tencel Siro-fil (196.7 dtex)				160	0.345
	14		Bamboo spun (196.7 dtex)				161	0.349
	15		Hi-multi PET (83.3 dtex)				133	0.301

**Table 4 materials-14-06205-t004:** Physical properties of the fabric specimens.

Group	Fabric Specimen No.	Pore Diameter D (µm)		Moisture Vapor Permeability				Thermal Conductivity		Absorption Rate R (%)
				Water Vapor Transmission Rate WVTR (g/m^2^·h)		Moisture Vapor Resistance R_ef_ (m^2^·Pa/W)		K (m^2^·Pa/W)		
		Mean	Dev.	Mean	Dev.	Mean	Dev.	Mean	Dev.(10^−3^)	Mean
A	1	3.82	0.100	425.4	11.1	1.91	0.103	0.0445	1.66	28.3
	2	3.19	0.088	413.2	10.0	2.80	0.109	0.0458	1.67	27.2
	3	2.85	0.112	410.6	9.2	3.42	0.118	0.0489	1.64	26.4
	4	2.98	0.110	430.2	11.1	2.91	0.101	0.0473	1.49	30.2
	5	3.56	0.124	385.5	10.0	1.74	0.107	0.0423	1.74	26.3
B	6	3.20	0.104	348.2	10.1	1.56	0.101	0.0439	1.57	30.2
	7	2.85	0.115	340.5	10.0	2.85	0.109	0.0437	1.67	28.5
	8	2.30	0.096	337.6	7.1	3.24	0.107	0.0481	1.58	28.1
	9	2.70	0.121	375.4	9.0	2.88	0.101	0.0453	1.67	32.4
	10	3.11	0.116	326.2	9.1	1.34	0.109	0.0395	1.64	26.2
C	11	3.45	0.100	369.8	11.0	2.12	0.118	0.0453	1.80	34.4
	12	2.95	0.116	358.6	11.1	2.43	0.103	0.0461	1.81	34.2
	13	2.55	0.106	355.4	10.3	3.32	0.101	0.0478	1.80	34.0
	14	2.76	0.094	378.5	9.1	2.86	0.118	0.0472	1.54	36.7
	15	3.25	0.111	337.7	11.2	1.78	0.101	0.0421	1.45	32.1

Note: dev = max − min.

**Table 5 materials-14-06205-t005:** ANOVA analysis of the fabric physical properties.

Physical Properties		F-Value(F_0_)	F(4, 20, 0.95)	*p*-Value
Pore diameter	Group A	318.0	2.87	8.57 × 10^−18^ (*p* < 0.05)
	Group B	221.9	2.87	2.91 × 10^−16^ (*p* < 0.05)
	Group C	279.3	2.87	3.06 × 10^−17^ (*p* < 0.05)
WVTR	Group A	83.7	2.87	3.36 × 10^−12^ (*p* < 0.05)
	Group B	129.7	2.87	5.29 × 10^−14^ (*p* < 0.05)
	Group C	67.4	2.87	2.54 × 10^−11^ (*p* < 0.05)
Ref	Group A	1119.5	2.87	3.31 × 10^−23^ (*p* < 0.05)
	Group B	1971.7	2.87	1.18 × 10^−25^ (*p* < 0.05)
	Group C	872.7	2.87	3.94 × 10^−22^ (*p* < 0.05)
K	Group A	64.4	2.87	3.90 × 10^−11^ (*p* < 0.05)
	Group B	124.8	2.87	7.66 × 10^−14^ (*p* < 0.05)
	Group C	54.7	2.87	1.72 × 10^−10^ (*p* < 0.05)

**Table 6 materials-14-06205-t006:** SEM images of cross-sections (×500) and surfaces (×150) of the yarns and optical microscopy (×300) [53].

Yarn Specimen No	Yarns	SEM (Cross-Section)	Optical Microscopy (Cross-Section)	SEM (Surface)
(1)	PP/Tencel Sheath/core	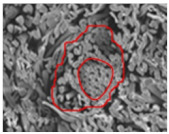	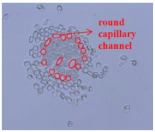	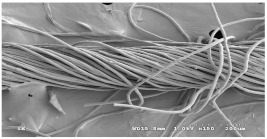
(2)	PET/Tencel Siro-fil	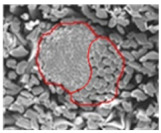	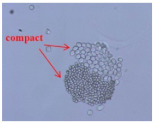	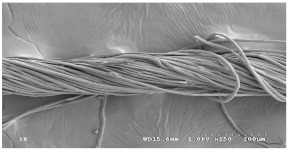
(3)	Tencel Siro-spun	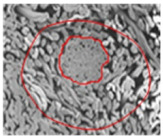	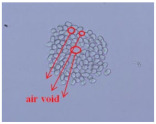	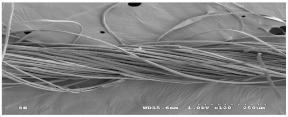
(4)	Coolmax/ Tencel Sheath/core	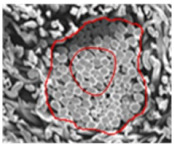	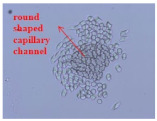	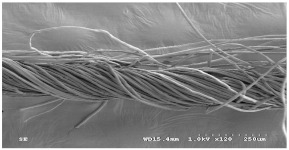
(5)	Coolmax/ Bamboo Spunyarn	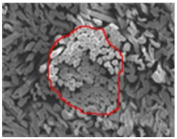	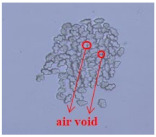	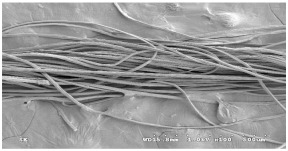
(6)	PET/ Tencel Siro-fil	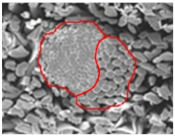	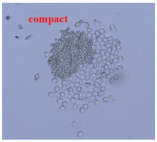	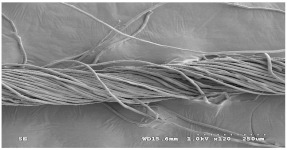
(7)	Bamboo Spunyarn	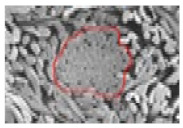	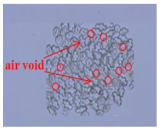	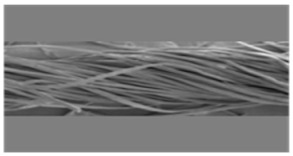
(8)	Hi-multi PET 75d/144f	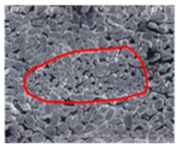	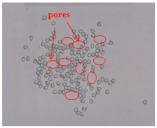	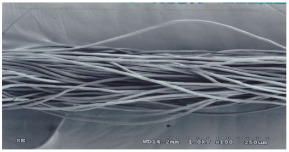
(9)	PP DTY 100d/48f	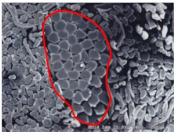	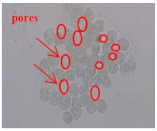	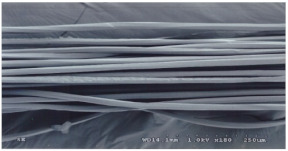

**Table 7 materials-14-06205-t007:** Correlation coefficient between each parameter of the wear comfort characteristics of the fabrics.

	Pore Diameter (µm)	Water Vapor Transmission Rate (g/m^2^·h)	Moisture Vapor Resistance (m^2^·Pa/W)	Thermal Conductivity (m·Pa/W)	Absorption Rate (%)
Pore diameter (D)	1				
Water vapor transmission rate(WVTR)	0.354	1			
Moisture vapor resistance (R_ef_)	−0.734 ^a^	0.264	1		
Thermalconductivity (K)	−0.545 ^b^	0.412 ^b^	0.872 ^a^	1	
Absorptionrate (A)	−0.226	−0.159	0.161	0.301	1

Note: ^a^ significant at the 0.01 level; ^b^ significant at the 0.05 level.

## Data Availability

The data presented in this study are available on request from the corresponding author.

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
