# Peer review of "Moisture Vapor Permeability and Thermal Wear Comfort of Ecofriendly Fiber-Embedded Woven Fabrics for High-Performance Clothing"

_materials, 2021, doi:10.3390/ma14206205_

Round 1

Reviewer 1 Report

All Introduction is written in one paragraph. That should be corrected.

In 2.1. section, the yarns' structure is described not enough clearly.

In Table 2, linear density of almost weft yarns is 196.7 dtex, except of Hi-multi PET (83.3 dtex). Is it correct?

It is not selected, what balance was used for samples' weight.

The first paragraph of section 3.1. is not clear: 329 row - maybe "in Table 4"; there is no deviation in Table 3 (337 row), etc.

The correlation coefficients 0.73 and 0.87 are not high, so, author cannot say that the dependencies are strong.

All Conclusions are written in one paragraph. That should be corrected.

Author Response

Please find the files attached.

Reviewer 2 Report

  1. The authors should present in the introduction the criteria and their quantitative values that correspond to environmental materials.
    2. Justify why polypropylene and other synthetic polymers are classified as environmental materials? This is important given the title of the article.
    3. Table 1 supplements the data on the period of biological decay in nature for the presented environmental materials.
    4. Line 211-212 "A fabric specimen with 47 mm in diameter was placed in the specimen holder shown in Figure 2". This is not shown in Fig. 2.
    5. Line 222-228 repeats the information from the introduction.
    6. Formula 3 requires quality improvement, the designations and their deciphers must be identical.
    7. Formula 5 requires quality improvement.
    8. Table 1 has information in mm, Fig.3a-in mm, formula 5 in cm, tab.3 in m. The authors need to provide data in meters in all sections for the accuracy of the information.
    9. The quality of Fig. 4 requires improvement.
    10. Figure 5 repeats the information from column 3 in tab.3; Fig. 7 repeats column 4 from Tab.3; Fig. 8 repeats column 2 of tube. 3; Fig.9 repeats column 6 from tab.3.
    11. Figures 9 and 10 require improvement. The authors should present and analyze the level of reliability of the established functions and the confidence of the correlation of points on the graph .
    12. The analysis of the submitted conclusion requires the re-submission of the article after completion

Author Response

Please find the files attached.

Round 2

Reviewer 1 Report

I accept all the corrections.

Reviewer 2 Report

The units of measurement of thermal conductivity should be corrected in Tab.5 and Fig.10 to (W/mK).

The article can be published.